# Influence of Upper Footwear Material Properties on Foot Skin Temperature, Humidity and Perceived Comfort of Older Individuals

**DOI:** 10.3390/ijerph191710861

**Published:** 2022-08-31

**Authors:** Pui-Ling Li, Kit-Lun Yick, Joanne Yip, Sun-Pui Ng

**Affiliations:** 1School of Fashion and Textiles, The Hong Kong Polytechnic University, Hung Hom, Hong Kong SAR, China; 2Laboratory for Artificial Intelligence in Design, Hong Kong Science Park, New Territories, Hong Kong SAR, China; 3School of Professional Education and Executive Development, The Hong Kong Polytechnic University, Hung Hom, Hong Kong SAR, China

**Keywords:** older adults, upper footwear materials, in-shoe skin temperature and relative humidity, foot skin temperature distribution, dorsal and plantar, thermography

## Abstract

Studying the in-shoe microclimate of older individuals is important for enhancing their foot comfort and preventing foot diseases. However, there is a lack of scientific work that explores the thermo-physiological wear comfort of older individuals with different footwear. This study aims to examine the effects of upper footwear materials on changes and distributions in the foot skin temperature and relative humidity for older individuals. Forty older individuals are recruited to perform sitting and walking activities under four experimental conditions in a conditioning chamber. The findings indicate that footwear upper constructed of highly permeable mesh fabric with large air holes shows fewer changes in foot skin temperature (ranging from 1.3 to 3.3 °C) and relative humidity (ranging from −13.3 to 5.7%) throughout the entire foot during dynamic walking, as well as higher subjective ratings on perceived thermal comfort when compared to footwear made of synthetic leather and composite layers. The findings serve to enhance current understanding of designing footwear with optimum comfort for older adults.

## 1. Introduction

Although the feet only account for 7% of the entire body surface area, they function as an important thermal radiator to thermally regulate the human body [1,2]. Footwear plays an important role in insulating the feet in daily life activities, and acts as a barrier in the transfer of heat and moisture between the skin and the external environment [3]. When different levels of activities are carried out, heat could be transferred within the footwear in the forms of convection, radiation, conduction, and latent heat transfer. Convection, radiation, and conduction transfer are mainly caused by the temperature difference between the surface of the skin of the foot and the environment as a form of dry heat transfer, whereas latent heat transfer refers to the water vapor pressure between the surface of the skin of the foot and the environment by means of moisture transmission [4,5]. Therefore, assessing the foot skin temperature and humidity is paramount for evaluating the in-shoe microclimate.

With the increasing prioritization of healthy and active lifestyles, footwear manufacturers today not only focus on fit and design, but also the wear comfort of footwear. The rapid growth of the footwear market has brought about significant challenges to the industry on preserving an optimum footwear microclimate, which is influenced by the surrounding temperature and humidity, shoe fit, foot movement, sweat production, and evaporation, as well as the in-shoe air movement or exchange. The overall in-shoe microclimate therefore largely depends on the construction of the footwear and the air enclosed and bounded by the footwear materials [1,6,7]. The amount of sweat produced by the foot enclosed by footwear also varies according to the thermal stress, exercise load, personal physiological characteristics, body heat storage, and footwear materials [8,9,10]. Foot sweat production during exercise could be associated with the type of footwear worn, which creates a high temperature and humidity environment in the footwear or slight hidromeiosis, where the skin is excessively hydrated due to the imbibition of sweat [8,11].

As the choice of footwear material has a major influence on the production and dissipation of heat and moisture throughout the entire foot, a large quantity of research work has been carried out on thermoregulatory responses and their association with footwear design and materials. Wrobel et al. (2014) carried out a study on insole materials and found that increased temperatures at the forefoot and midfoot can be significantly reduced by using intervention insoles made of different combinations of polyethylene, polypropylene, ethylene vinyl acetate, and polyurethane materials [12]. Irzmańska (2016) also found that protective footwear for firefighters made of leather causes a higher foot skin temperature compared to footwear made of nitrile rubber, which results in a higher relative humidity (RH) inside the footwear [13]. Sweat production of young female runners who run in their bare feet was found to be 22% higher than that of wearing shoes with a mesh upper. Nevertheless, the foot skin temperature increased in the shod condition during low-density running [10]. The use of running shoes with a less permeable mesh upper construction results in a high in-shoe temperature and humidity, which can be perceived to be more uncomfortable during running in comparison to wearing shoes with a more permeable mesh upper [14]. Footwear made of genuine leather has a lower absolute in-shoe humidity than that made of synthetic leather for all parts of the foot [1]. However, previous studies have only focused on the changes in foot temperature and humidity of young adults but neglected the specific footwear needs of older adults. The response of cutaneous thermoreceptors to heat and thermoregulatory performance both deteriorate with age. Hence, older adults are less sensitive to thermal responses, especially when performing physical activities in higher-temperature environments [15,16]. In comparison to young adults, older adults store more body heat when exposed to heat load during both passive heat exposure and physical activity [17,18,19]. They also have lower local and body sweat rates, and a delayed core temperature threshold for the onset of sweating, thus leading to a higher core temperature [19,20] and lower peripheral blood flow, which limit the transfer rate of heat to the surface of the skin with smaller skin temperature increases and less efficiency in sweating and heat evaporation [16]. During daily activities, the combination of high temperature and perspiration inside footwear not only leads to perceived foot discomfort, but also nurtures the growth of microorganisms, such as fungi and bacteria, and results in high risks of infections, chronic foot diseases, and tinea pedis [21,22]. However, there is a scarcity of scientific work that has examined the thermal properties of footwear upper materials to improve the thermo-physiological wear comfort of footwear that is closely related to the specific design, footwear ventilation, intended end-use, and even age and physiological response of the wearers. To enhance the wear comfort for optimal foot protection and health of older adults, especially those aged over 60 years old, attention should be given to the in-shoe skin temperature distribution and humidity of the entire foot, which are related to the thermal properties and behavior of the footwear materials.

The aim of this study is therefore to examine the effects of upper footwear materials on the RH and temperature changes and distributions of the skin of the foot of older individuals, including synthetic leather, mesh fabric, and multi-layered fabrics that have different performance in terms of permeability and thermal comfort. Since the properties of footwear materials can affect the temperature and level of humidity of the shoe microclimate and thermal comfort [23], the thermal properties of upper footwear materials are systematically investigated. To provide a comprehensive understanding of the thermoregulation ability of the foot and that of older individuals, the skin temperature and humidity distributions across the dorsal and plantar of the foot are analyzed. Additionally, the impact of the thermal environment in footwear with sitting and walking activities on subjective sensation is examined. It is anticipated that a higher rate of sweat secretion of the dorsal of the foot towards heat loss with different upper footwear materials would have crucial impacts on the in-shoe microclimate. The findings of this study are important for improving footwear design and selection of upper footwear materials for the elderly, especially those with foot problems and ulcers, to provide better in-shoe microclimate and thermal comfort for their daily usage and prevent foot diseases commonly found among the elderly.

## 2. Materials and Methods

### 2.1. Participants

In total, 40 older individuals were recruited for the study. Twenty-five were female who were between 63 and 79 years old (mean: 70; SD: 4.1), whereas 15 were male who were between 63 and 82 years old (mean: 70; SD: 5.6). The feet of all of the participants had no serious problems (e.g., the angle of the hallux valgus is less than 30 degrees), the participants were not taking any prescription medication, and had no history of cardiovascular disease, skin disease, prediabetes, wounds, varicose veins, ankle edema, or foot hyperhidrosis. Their cognition and verbal comprehension were within normal range of giving fair ratings for subjective sensations. They were also able to walk independently for a long time (e.g., at least for 2 h). The body mass index and foot size of the older women ranged from 17.5 to 28.8 kg/m^2^ (mean: 22.5; SD: 3.1) and EU 37 to 40, respectively. Those of the older men ranged from 19.0 to 29.0 kg/m^2^ (mean: 23.5; SD: 2.6) and EU 37 to 43, respectively. All of the participants gave written informed consent before participating in the study. The experiment was approved by the Human Subjects Ethics Sub-Committee at The Hong Kong Polytechnic University prior to starting the study.

### 2.2. Experimental Conditions

The experiment consisted of four conditions: bare feet—Condition A, and wearing three different types of footwear—Footwear Conditions B, C, and D (Figure 1). All of the footwear used had standard insoles made of ethylene vinyl acetate (EVA) and a layer of leather upper (with a thickness of 8 mm), which had a hardness of Shore A30 and coefficients of static and kinetic friction of 0.50 and 0.36, respectively [24]. The soles of Footwear Conditions B and C were made of EVA with a hardness of Shore A50 and Shore 45, respectively, whereas the sole of Footwear Condition D was made of rubber (Shore A65). An Artec Eva 3D handheld scanner (Artec Group, Luxembourg) and Geomagic Studio 2012 software were used to scan the footwear and then measure the corresponding area covered by the shoe upper material. Footwear Conditions B and C, which were leather and mesh sports shoes, respectively, had a similar shape, design, and covered area, whereas Footwear Condition D, which was a closed-toe slipper, had a smaller covered area (see Figure 1). The material tests with specific test standards to characterize the thermal properties of the upper material are listed in Table 1. Thermal conductivity and insulation tests (KES-F7 Thermo Labo II) were carried out to measure the ability of the materials to conduct heat and the amount of heat loss [25], and the air resistance (ASTM D737) and water vapor transmission (ASTM E96) rates were also measured to evaluate the resistance of air flow and the passage of water vapor through the materials. Footwear Condition B, which was made of synthetic leather, showed the lowest air and water vapor permeabilities, whereas Footwear Condition C, which was made of mesh fabric, had the highest air and water vapor permeabilities. Footwear Condition D, which had multiple layers of fabrics, had the lowest thermal conductivity but the highest thermal insulation.

### 2.3. Experimental Protocol

The experiment was carried out in a conditioning chamber and the ambient conditions were controlled at a setting of a temperature of 22 ± 1 °C, humidity of 60 ± 5%, and air velocity of 0.24 ± 0.01 m/s [26,27]. During the experiment, all of the participants wore standard cotton sportswear with a clothing insulation equal to 0.3 clo, including a pair of trousers and a long-sleeved T-shirt. The participants were required to perform sitting and walking activities in all of the footwear without socks as well as in their bare feet. For thermalization purposes, the participants were asked to sit in the conditioning chamber for 30 min prior to the start of the experiment. The sitting test was 20 min in length, and the walking test was 30 min and conducted on a treadmill at a speed of 3 km/h (Figure 2). Their footwear were weighed before and after each trial to determine the changes in weight due to sweat accumulation [28]. A total of 6 humidity sensors, which are 17.35 mm in diameter and 5.89 mm in thickness (Thermocron HC, DS1923, OnSolution), were also embedded into the standard insoles and attached to the dorsal side of the right foot to record the RH of the shod conditions only during the experiment. The temperature of the skin of the foot was recorded by using an infrared camera (FLIR T420bx, FLIR^®^ Systems, Inc., Wilsonville, OR, USA) with a thermal sensitivity of <0.045 °C at 30 °C for all of the experimental conditions. Thermal images of the foot were taken twice: (1) before each test, and (2) after the test (within 1–2 min after the shoes were removed), from the dorsal and plantar sides at a distance of 0.8 m [29,30,31]. The camera was kept perpendicular to the areas of interest. To minimize the influence of the infrared radiation reflected from the body of the participants and the treadmill, anti-reflective black fabric for blocking the reflected infrared radiation was placed in front of the subjects or under their feet [32]. At the end of each trial, the participants were to rate the foot conditions based on three subjective scales, including the perceived thermal comfort, moisture sensation, and overall comfort (Figure 3). The sequence of the experimental conditions was randomized for each subject to reduce possible order effects. To allow the foot to revert back to its initial temperature that was recorded during thermalization, the participants were allowed to rest for 20 and 40 min after the sitting and walking tasks, respectively, and given cold water and alcohol to apply to their feet so as to speed up the recovery process. The average recovery time for sitting and walking was around 15 and 35 min, respectively. The temperature of the foot of each subject was monitored before starting the testing in the other conditions.

### 2.4. Statistical Analysis

The thermal images were processed by using Matlab R2021a. In this study, only the dominant foot (right foot) was used in the analysis [33,34]. The right foot of all of the subjects was then scaled from 0 to 100%, starting from the toes for all of the images (Figure 4a). Eleven temperature points were extracted in a percentage along the line that located in the middle of the dorsal and plantar foot (Figure 4a) to represent the temperature distribution of the whole foot in Figure 5, and the temperature measurements obtained from the plantar (TP1-TP4) and the dorsal (TD1-TD3) sides of the foot were inputted into the Statistical Package for the Social Sciences (SPSS) Version 26.0 (SPSS Inc., Chicago, IL, USA). The RH data were also extracted from the six loggers placed on the right foot for analysis (Figure 4b). All the data was approximately normally distributed. A repeated measures analysis of variance (ANOVA) was used to determine the effects of the type of footwear, activity, activity length (before versus after), and gender of the participants on the changes in temperature and RH. Bonferroni-adjusted post hoc tests were subsequently conducted to compare the main effects among the different conditions. The relationships among the perceived heat, humidity, and comfort were analyzed by using Spearman’s correlation coefficients, and the Kruskal–Wallis one-way ANOVA was also used to compare the perceived differences among the experimental conditions. The level of significance was set at 0.05.

## 3. Results

### 3.1. Changes and Distribution in Foot Skin Temperature

Throughout the length and level of activity (sitting and dynamic walking), the older adults showed significant changes in their foot skin temperature for most of the measured locations in the dorsal (TD1-TD3) and plantar (TP1-TP4) sides of the foot, except for TD1 of the dorsal side during the course of the activity (see Table 2).

After 20 min of sitting and 30 min of walking, the average foot skin temperature distribution of the dorsal side of all the feet was similar among all of the footwear conditions, as shown in Figure 5, in which the temperature is the lowest at the dorsal side of the toes and starts to increase at the dorsal side of the metatarsal area. The highest skin temperature was at the instep, and then the temperature slightly decreased with distance from the instep. The foot skin temperature distribution of the plantar side of the foot differed from that of the dorsal side, but the trends were similar among all of the experimental conditions. During sitting, the foot skin temperature was lower and started to increase beginning at the distal phalanges and was the highest at the mid-point of the plantar of the foot (arch), but started to drop and was lower at the plantar heel. During walking, the foot skin temperature increased starting from the distal phalanges and was higher at the center of the plantar of the foot (area from the plantar side of the metatarsal heads to mid-heel), and started to slightly decline at the proximal plantar heel. The overall lowest foot skin temperature distribution was found with Condition A (barefoot) for both the dorsal and plantar sides of the foot after sitting and walking. The highest was observed with Footwear Condition B for both the dorsal and plantar sides of the foot during sitting. Footwear Conditions B and D showed a similar as well as the highest foot skin temperature distribution during walking. Footwear Condition D showed a slightly higher temperature distribution at the forefoot area for both the dorsal and plantar sides of the foot, whereas Footwear Condition B showed a slightly higher temperature distribution at the instep for the dorsal and the plantar heel.

**Figure 5 ijerph-19-10861-f005:**
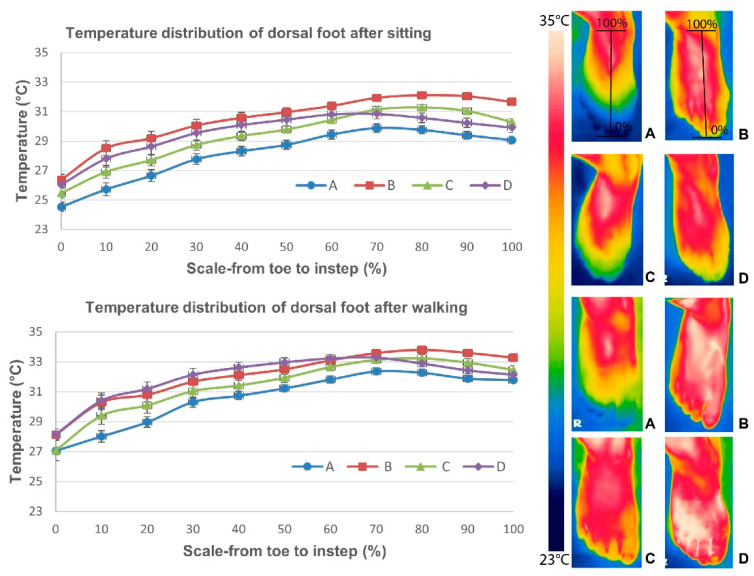
Temperature distribution (mean) of dorsal and plantar sides of foot based on four experimental conditions after sitting and walking, where A: barefoot, B: leather sports shoes, C: mesh sports shoes, and D: closed-toe slipper.

According to Figure 6, the older adults in this study tended to experience a reduction in all of the temperature points at the dorsal of the foot in most of the conditions during sitting, especially with Condition A, but a slight increase was found at the plantar of the foot for all of the footwear conditions, especially Footwear Condition B. The largest mean temperature difference for dorsal foot between Condition A (barefoot) and three Footwear Conditions B, C, and D was found at TD1 (toes), with 2.8 °C, 1.2 °C, and 1.8 °C, respectively, whereas for the plantar foot it was found at TP4 (heel), with 3.1 °C, 2.3 °C, and 2.5 °C, respectively. During walking, the foot skin temperature increased in all of the experimental conditions, but the lowest increase was found in Condition A for all of the temperature points of the dorsal and plantar sides of the foot, and the highest increase was found with Footwear Condition D. A significant increase was found in Footwear Conditions B and D in comparison to Condition A at TD1 and TD3, whereas the foot skin temperature increment in Footwear Condition B was also significantly higher than that in Condition A at TD2, and no significant difference was found between Condition A and Footwear Condition C (see Table 2). The largest mean temperature difference for the dorsal foot between Condition A (barefoot) and the three Footwear Conditions B, C, and D was found at TD1 (toes), with 2.3 °C, 1.4 °C, and 2.4 °C, respectively. For the plantar of the foot, the changes in foot skin temperature were significantly higher for all of the footwear conditions in comparison to Condition A at all of the temperature points, except TP1 between Condition A and Footwear Condition C. The largest mean temperature difference between Condition A (barefoot) and Footwear Condition B was found at TP1 (toes), with 2.4 °C, whereas between Condition A and Footwear Conditions C and D it was found at TP3 (arch), with 2.0 °C and 2.5 °C, respectively. No gender difference was found in terms of temperature changes in all the studied points.

### 3.2. Changes in Foot Skin Relative Humidity

During the length of the activity, the RH of the skin of the foot changed significantly at all of the points, except for HP3, whereas significant changes were only found at HP1, HP3, and HP4 in terms of level of activity (see Table 2).

During sitting, the older adults showed a similar and larger increment in the RH of the skin of the plantar side of their foot with Footwear Conditions B and D, whereas the former caused the highest change in RH at the dorsal side (see Figure 7). During walking, the RH of the plantar side tended to decrease with most of the footwear conditions, especially Footwear Condition C at HP3 (plantar arch), whereas that of the dorsal side tended to largely increase with Footwear Condition B. According to Table 2, the changes in the RH with Footwear Conditions B and D were significantly higher than Footwear Condition C at HP1 (plantar side of toes), whereas Footwear Condition C showed a significant difference at HP3 in comparison to Footwear Condition B. Footwear Condition D showed a significant increase at HP4 (plantar heel) in comparison to Footwear Condition C. A gender difference was only found at HP3, where the male subjects experienced a greater change in RH with a mean difference of 5.28%.

### 3.3. Sweat Accumulation in Footwear Conditions

With reference to Figure 8 and Table 2, a significant difference was found in the amount of sweat accumulated in the footwear for the length of the activity. Footwear Condition B was observed to cause a significant increase in the amount of accumulated sweat in comparison to Footwear Conditions C and D during sitting and walking, whereas Footwear Condition C resulted in the least amount of accumulated sweat. A gender difference was found for the amount of sweat accumulated, as the male subjects accumulated more sweat.

### 3.4. Subjective Perceptions

Significant correlations were only found between the subjectively perceived heat and moisture (*p* < 0.001), where the Spearman’s correlation coefficients between heat and comfort, moisture and comfort, and heat and moisture were −0.012, −0.079, and 0.412, respectively. In comparison to Condition A, significant differences were found for the perceived heat in Footwear Conditions B, C, and D (*p* < 0.001); for perceived moisture in Footwear Conditions B (*p* = 0.001) and D (*p* < 0.001); and for perceived comfort in Footwear Condition C (*p* = 0.002). The subjects felt that Footwear Conditions B and D during sitting and walking made their feet feel more warm and moist, whereas Footwear Condition C felt more comfortable (not as warm and moist) to them (Figure 9). However, significant differences in the subjective perceptions among the different footwear conditions were only found for the perceived heat between Footwear Conditions C and D (*p* = 0.042).

## 4. Discussion

In comparison to Condition A (barefoot), temperature increased with the use of shoes, which ranged from 0.7–2.8 °C for dorsal foot and 1.3–3.1 °C for plantar foot. The temperature increases in all of the footwear conditions were significantly higher for most of the temperature points on the plantar of the foot, especially during gait. These findings are supported by the study of Priego Quesada (2015) that shod running elicited higher temperatures than barefoot running (0.5–2.2 °C for forefoot and 0.9–2.4 °C for midfoot) [35], which might be due to the insulation of footwear and generation of heat in the midsole and the higher friction between the foot and the sole of the footwear due to slipping, along with the presence of environmental variables, such as air temperature and solar radiation, which would result in higher temperatures [35,36]. During walking, the skin temperature changes at the plantar side of the foot were higher at P2 (highest temperature change of 4 ± 0.23 °C at Footwear Condition D) and P4 (4 ± 0.21 °C at Footwear Condition B), which are on the plantar side of the metatarsal heads and heel, respectively, and would be mostly in direct contact with the ground during gait. This result is consistent with a previous study by Shimazaki and Murata (2015) in that the increases in the skin temperature of young males were higher in the plantar side of the metatarsal heads and the heel (3.5 °C) when wearing shoes with a porous material at a gait speed of 3km/h, whereas those in the regions with less contact with the ground, including the plantar side of the toes (1.6 °C ) and arch (1.2 °C) and dorsal side of the foot, were relatively lower [36]. This phenomenon is due to the occurrence of strong contact forces when the body weight is solely placed on the heels at initial contact with the ground and the forces are concentrated again over a small region of the forefoot in the late stance during walking. The higher and repeated contact load/force in these areas produces heat energy, which is converted from mechanical energy. The absorption of the contact force on the skin itself through the shoe cushion leads to temperature increases [36], which indicates that the materials used for the plantar side of the metatarsal heads and heel at the insoles should have high thermal conductivity, good air permeability, and lower friction to facilitate heat transfer and reduce heat generation. However, the increase in the foot temperature after walking in this study was lower (4 °C) in comparison to the results of a study by Shimazaki et al. (2016), who targeted young adults (around 6 °C) [1]. The results of both studies during sitting are similar (less than 1 °C). The foot skin temperature of older adults recorded in this study was also lower in comparison to the foot temperature (mean) of the young adults (around 35 °C) in Reddy et al. (2017) [37], in which the highest temperature points for the dorsal and plantar sides of the foot with footwear after walking were 33.8 °C and 32.6 °C, respectively. These results are supported by the findings in Inbar et al. (2004), in which older adults had a more slow and smaller increase in foot skin temperature during exercise in comparison to young adults [16].

In comparison to the plantar side of the foot, the older adults experienced a smaller increase/decline in foot temperature at the dorsal side of the foot, especially with Footwear Conditions B, C, and D, whereas the lowest increments were shown with Footwear C. This might be because the heat generated and accumulated from the body can easily be dissipated from the dorsal rather than the plantar side of the foot to the external environment by different mechanisms of heat loss, such as evaporation through footwear material, which has high water vapor permeability and openings, convection caused by ventilation in the footwear through the surface of the footwear components with high air permeability and openings, and radiation from the skin surface [4]. The heat loss from the dorsal side of the foot is largely affected by the convective heat release, which is related to the motion of the feet. The release of convective heat occurs in complex ways. When the temperature of the skin of the foot is higher than the ambient temperature, the heat is transferred to the footwear surface and the ambient environment through convection. The heat is dissipated from the footwear by convection when the surface of the footwear warms up [1]. In addition, the sweat glands on the surface of the plantar side of the foot are mainly stimulated through psychological stress or to a lesser extent, by changes in body heat storage, such as those due to emotions, anxiety, or friction, between the contact surface and the skin. On the other hand, the sweat glands on the surface of the dorsal side of the foot respond more to thermal stimuli than psychogenic influences [8]. Since the skin of the plantar side of the foot is not responsible for thermoregulatory sweating and plantar sweating would only increase invariably when thermal strain and exercise loading are elevated [9], the dorsal side of the foot becomes more important for thermoregulatory sweating and tends to produce more sweat and local heat caused by air restriction, which increase the tendency to sweat in that area and also contribute to the cooling effects of the temperature of the skin [1,28]. Smith et al. (2013) found that the dorsal side of the foot produces 65–70% of the sweat discharged from the skin of the dorsal of each foot [8], whereas only around 30% is from the surface of the plantar of the foot under significant exercise (incremental cycling) [9]. The sweat secretion could be attributed to the eccrine sweat glands that are distributed over the palm and sole of the feet [38]. However, the conduction of heat that is accumulated under the plantar of the foot might be mainly taking place through the soles so that there is minimal heat loss [4]. The changes in the RH and amount of sweat accumulation due to gender differences are in agreement with findings in previous studies in that males sweat 2.2 times more than females in all areas of the feet [8]. Therefore, males are more sensitive to humidity in comparison to females [39].

In comparison to Footwear Condition C (highest increase of 3.3 °C in temperature and 8.4% in RH), the in-shoe microclimate in Footwear Conditions B and D tended to be warmer and more moist since the foot had the highest increment in skin temperature and RH for both the dorsal and plantar sides (highest increase of 4.0 °C in temperature for both conditions and 13.2% and 12.1% in RH, respectively) during both static (sitting) and dynamic (walking) activities. This might have been due to the non-porous leather upper of Footwear Condition B and multiple layers of fabric of Footwear Condition D with foam, which is thicker and reduces the air ventilation/flow along the skin and limits the heat loss pathways, thus trapping heat, and especially inhibiting convection and evaporation [40]. Eventually, these would result in a higher humidity and increased temperature, especially at the plantar side of the foot. The single-layer mesh fabric of Footwear Condition C with large air holes over its entire surface means that it had the highest air and water vapor permeabilities, which largely facilitated a higher rate of air exchange and bellows action for the transfer of heat from the footwear to the ambient environment, especially at the dorsal side. The narrower space between the foot and Footwear Condition C would also have led to larger pressure differences between the shoe and the environment, in which the forced airflow created helped to facilitate the bellows action for heat loss [7]. There was therefore no significant skin temperature difference between Condition A and Footwear Condition C. Although the skin of the foot experienced a larger increase in temperature in Footwear Conditions B and D (which had upper materials with high thermal insulation and low water vapor transmission), the increase was not significant when compared to Footwear Condition C. However, there was a significant increase in the RH. The composite materials and synthetic leather combined with foam and lining materials absorb moisture from the skin. Their contact surface retains the moisture and so the feet of the wearer remain damp. The increased heat in the footwear is then either dissipated to the ambient environment or trapped between the footwear and the skin. The higher temperature and RH found inside Footwear Conditions B and D might promote the growth of fungi and bacteria, which lead to higher incidence of tinea pedis and infectious diseases for the elderly, especially for those with diabetic-associated feet and foot ulcers. This is unlike Footwear Condition C, which used material that is thin with large air holes and good water vapor transmission to wick moisture away from the skin, thus resulting in greater perceived comfort and minor differences in the in-shoe skin temperature and humidity. This means that the air holes of the upper materials of footwear play an important role in determining the in-shoe microclimate. Air holes are therefore recommended for the instep, medial, and lateral sides of the vamp when designing footwear for older adults to enhance their thermal comfort. On the other hand, the overall temperature distribution of both the dorsal and plantar sides of the foot was the highest with Footwear Condition B during sitting, whereas the highest temperature distribution was found at the forefoot with Footwear Condition D and in the instep (dorsal) and plantar heel with Footwear Condition B during walking. These results are attributed to the poor air permeability of Footwear Condition B and the properties of low thermal conductivity with high insulation of Footwear Condition D so that heat is readily accumulated with limited transfer from inside of the footwear to the outside, leading to high foot temperature distribution in the dorsal side of the toes and forefoot areas. The slightly higher temperature distribution of the instep (dorsal) and plantar heels found in Footwear Condition B might also have been due to the larger covered area of the instep than in Footwear Condition D, and the thicker and less permeable tongue material than in Footwear Condition C, which reduces the convection and accumulates more heat during the gait at the heel area as well.

In this study, the in-shoe skin temperature and RH significantly changed during an activity itself and the length of the activity since more heat and sweat are produced and accumulated from the body during dynamic activities from the contact force, contact duration, contact frequency, etc. During the static activity, which was sitting in this study, the temperature of the skin at the dorsal and plantar sides of the foot tended to drop, or there were almost no changes even when the footwear was worn, whereas the in-shoe RH of the skin of the foot showed a trend of increase. In comparison to sitting, the skin temperature at both the dorsal and plantar sides of the foot was greatly increased during walking, but the increment of RH was reduced or even declined, except at HD1 (dorsal side of the metatarsal heads) with Footwear Conditions B and D, and HP5 (plantar side of the fourth and fifth metatarsal heads) with Footwear Condition D. This might have been due to the higher convective heat loss during walking through the increased air flow around the shoes and greater movement of the foot within the footwear, thus causing a pumping action, which forces air into the environment [7,14]. The increased ventilation would limit the increase in skin moisture accumulation and result in a reduction in the amount of accumulated moisture within the footwear [14,41]. The reduction of the in-shoe RH of the skin of the foot might also have been due to the suppression of the rate of sweating when wearing footwear under an increased exercise load [8]. The largest decline in the RH during walking was found at HP3 for all of the footwear conditions, especially Footwear Condition C (-13.3% RH), which might have been due to the hollow shape of the arch and the mesh fabric with large air holes, as the upper material would be directly influenced by the external environment and more easily wick moisture through the bellows action during more foot movement, such as the set air temperature and RH in the experimental chamber. The increase in the RH of the skin of the foot at HD1 with Footwear Conditions B and D during walking might have been due to the higher rate of sweating of the dorsal side of the foot. Furthermore, the less permeable and highly insulated upper materials (foam and multiple layers) were a contributing factor for the larger increase in RH, as there was a lower rate of transfer of moisture as opposed to the generation of moisture.

## 5. Conclusions

To conclude, the upper material properties of footwear play an important role in the in-shoe microclimate, affecting the dissipation of heat and moisture of the skin of the foot in response to the physical activity of older individuals. In this study, footwear materials with higher thermal insulation and lower air and water vapor permeabilities resulted in skin temperature increases (highest increase of 4.0 °C) and variations in the skin temperature distribution in both the dorsal and plantar sides of the foot of older adults, as well as higher humidity (highest increase of 13.2%) at the plantar side of the foot, especially during walking. Higher skin temperatures were found at the instep (33.8 °C) of the dorsal of the foot and in the areas of the metatarsal heads (32.6 °C) and heel (31.9 °C) of the plantar of the foot. Heat and sweat accumulated in the footwear, however, can easily dissipate from both the dorsal (highest increase of 1.6 °C and 7.2%) and plantar sides (highest increase of 3.3 °C and 8.4%) of the foot when the upper footwear material is made of highly permeable mesh fabric. The large air holes of mesh upper facilitates heat and moisture dissipation, which not only preserves the skin temperature and relative humidity throughout the entire foot, but also enhances the perceived thermal comfort and foot health for older people. The findings of this study provide fundamental information on improving the selection of materials when designing footwear for older people so that they experience greater thermal comfort, which could prevent adverse foot conditions.

## Figures and Tables

**Figure 1 ijerph-19-10861-f001:**
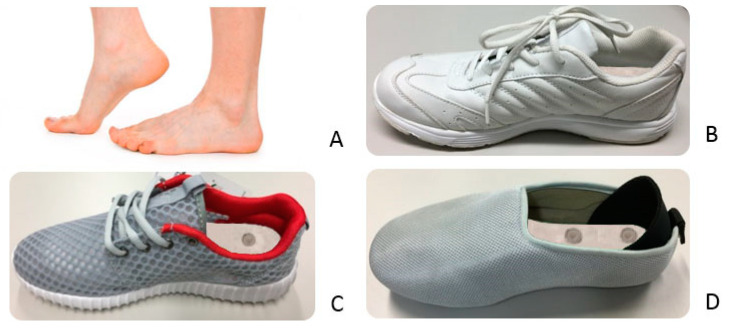
Four experimental conditions: Condition (**A**), barefoot; Footwear Condition (**B**), leather sports shoe; Footwear Condition (**C**), mesh sports shoe; and Footwear Condition (**D**), closed-toe slipper.

**Figure 2 ijerph-19-10861-f002:**
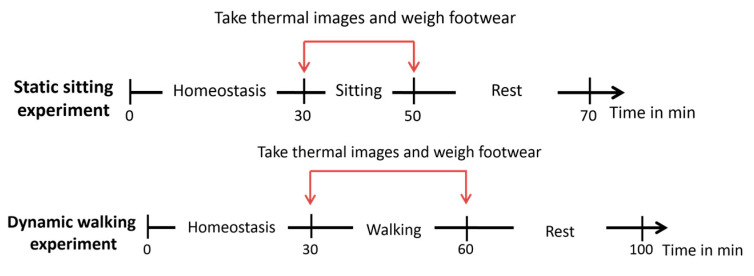
Experimental procedures.

**Figure 3 ijerph-19-10861-f003:**
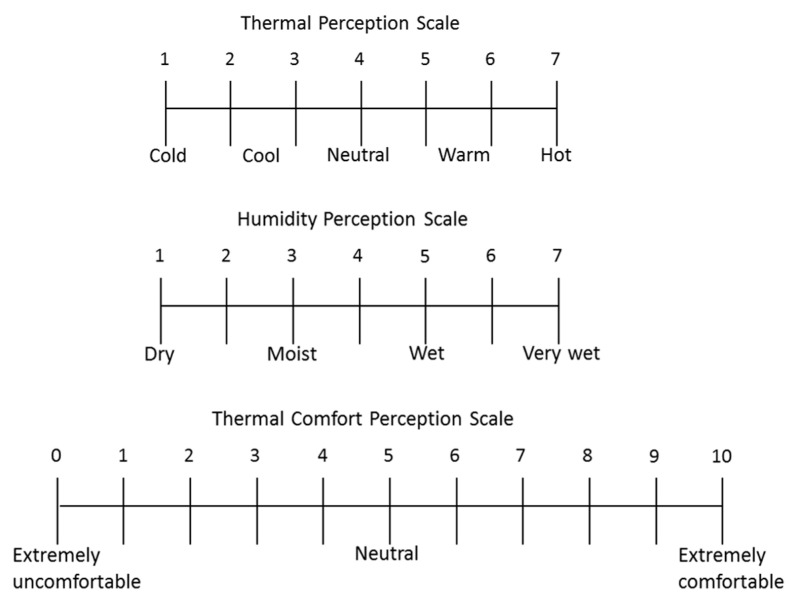
Rating scale of subjective sensations.

**Figure 4 ijerph-19-10861-f004:**
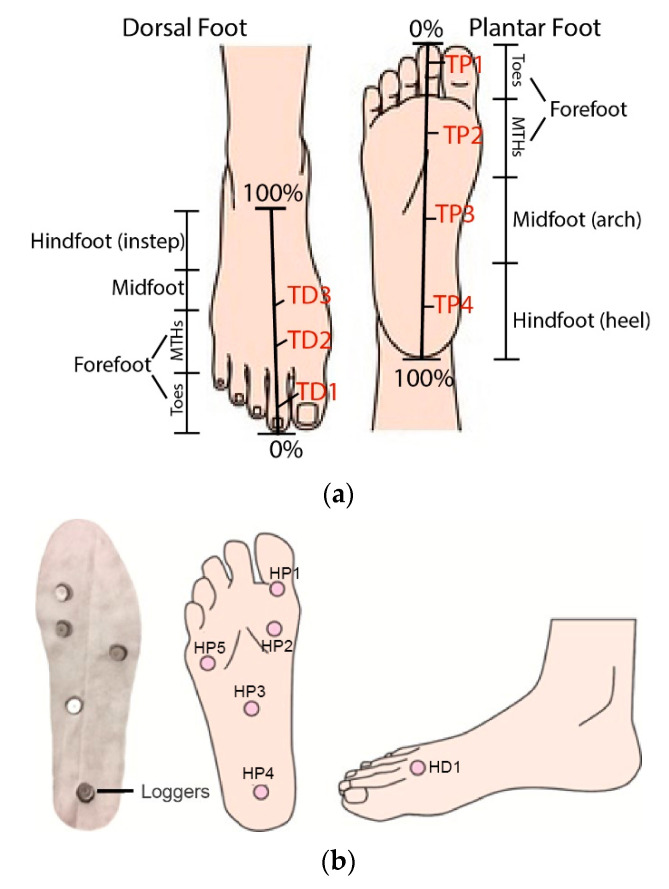
(**a**) Locations measured for temperature. (**b**) Locations measured for relative humidity.

**Figure 6 ijerph-19-10861-f006:**
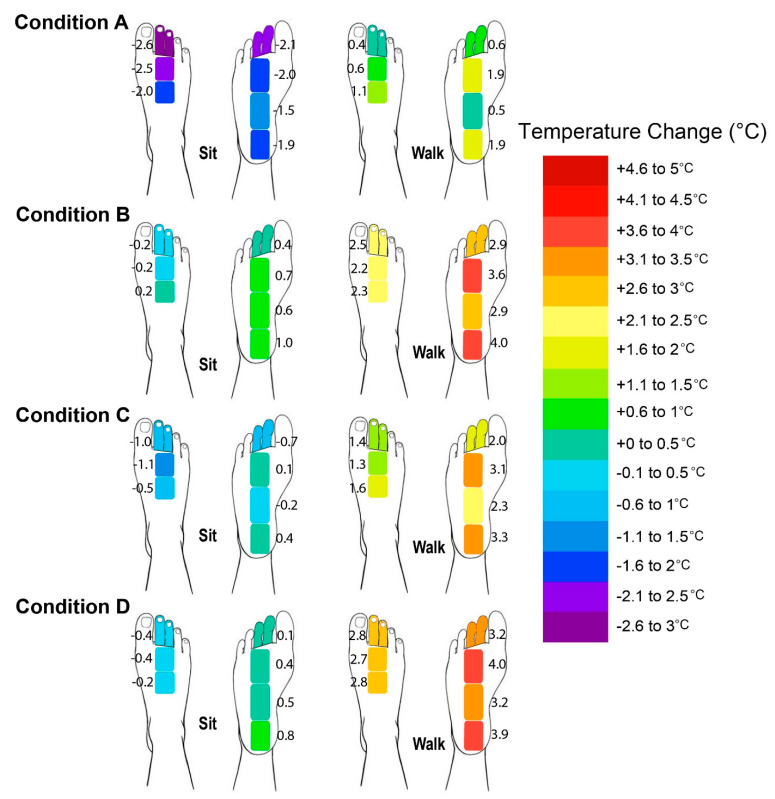
Changes in foot skin temperature of dorsal (TD1–TD3) and plantar sides (TP1–TP4) of foot based on four experimental conditions after sitting and walking, where A: barefoot, B: leather sports shoes, C: mesh sports shoes, and D: closed-toe slipper.

**Figure 7 ijerph-19-10861-f007:**
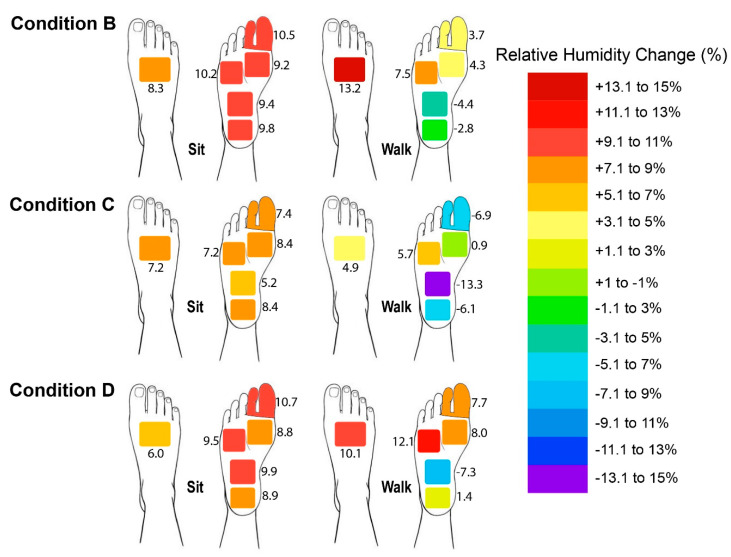
Changes in foot skin relative humidity at plantar (HP1–HP5) and dorsal (HD1) sides of foot after sitting and walking, where B: leather sports shoes, C: mesh sports shoes, and D: closed-toe slipper.

**Figure 8 ijerph-19-10861-f008:**
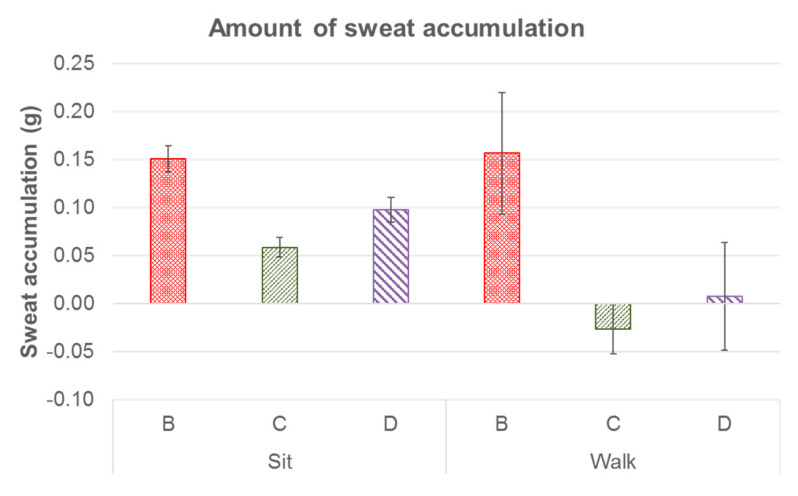
Amount of sweat accumulated for three footwear conditions during sitting and walking, where B: leather sports shoes, C: mesh sports shoes, and D: closed-toe slipper.

**Figure 9 ijerph-19-10861-f009:**
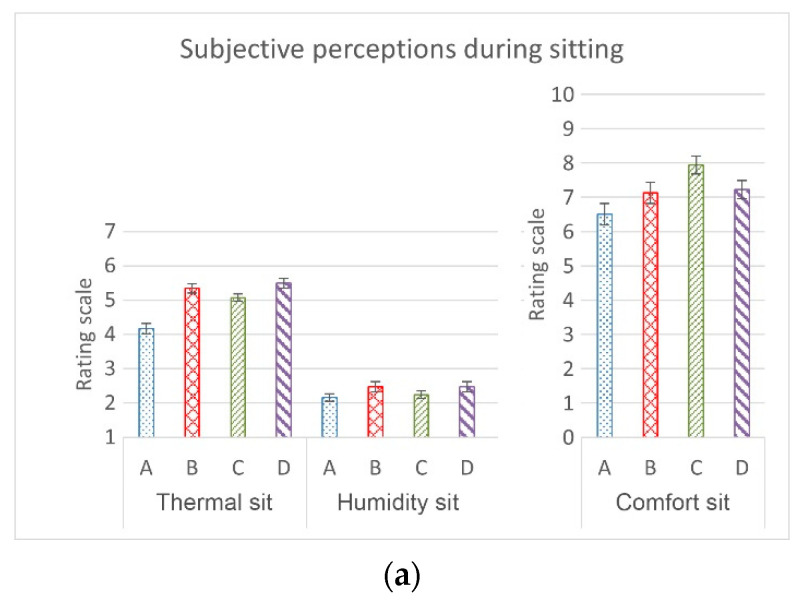
Subjective sensations with different footwear conditions during (**a**) sitting and (**b**) walking, where A: barefoot, B: leather sports shoes, C: mesh sports shoes, and D: closed-toe slipper.

**Table 1 ijerph-19-10861-t001:** Upper material properties of footwear conditions.

	Footwear Condition B	Footwear Condition C	Footwear Condition D
Type of fabric	Synthetic leather with foam and lining	Mesh fabric	A combination of woven, foam, and knit (lining) fabrics
Thickness (mm)	4.74	1.90	6.38
Covered Area—EU 37–43 (cm^2^)	498.84–505.24	494.26–505.64	441.33–445.07
Thermal conductivity KES-F7 (W/cm·K)	0.040	0.042	0.034
Thermal insulation KES-F7 -heat retention rate (%)	53.69	42.99	68.11
Air resistance KES F8 (kPa·s/m)	-	0.01	0.07
Water vapor transmission rate ASTM E96 (g/m^2^·h)	0.84	38.06	27.01

**Table 2 ijerph-19-10861-t002:** Summary of effects of different factors on changes in skin temperature, relative humidity, and sweat accumulation.

		Factor (*n* = 40)									
						Experimental Condition					
		Location	Length of Activity	Activity	Gender	A and B	A and C	A and D	B and C	B and D	C and D
**Temperature change**	**Dorsal side**	TD1	0.096	**0.005**	0.757	**<0.001**	1.000	**0.006**	0.140	1.000	1.000
		TD2	**0.033**	**0.002**	0.733	**0.002**	1.000	0.056	0.185	1.000	1.000
		TD3	**<0.001**	**<0.001**	0.996	**<0.001**	1.000	**0.003**	0.067	1.000	0.273
	**Plantar side**	TP1	**<0.001**	**<0.001**	0.483	**<0.001**	0.195	**0.001**	0.306	1.000	1.000
		TP2	**<0.001**	**<0.001**	0.737	**<0.001**	0.009	**<0.001**	1.000	1.000	1.000
		TP3	**<0.001**	**<0.001**	0.632	**<0.001**	<0.001	**<0.001**	1.000	1.000	1.000
		TP4	**<0.001**	**<0.001**	0.589	**<0.001**	<0.001	**<0.001**	1.000	0.357	1.000
		Location	Length of activity	Activity	Gender	A and B	A and C	A and D	B and C	B and D	C and D
**RH change**	**Plantar side**	HP1	**<0.001**	**<0.001**	0.124	N/A	N/A	N/A	**<0.001**	1.000	**<0.001**
		HP2	**<0.001**	0.359	0.616	N/A	N/A	N/A	0.257	1.000	0.149
		HP3	0.908	**<0.001**	**0.013**	N/A	N/A	N/A	**<0.001**	0.409	0.066
		HP4	**<0.001**	**<0.001**	0.176	N/A	N/A	N/A	0.197	0.132	**0.001**
		HP5	**<0.001**	0.923	0.459	N/A	N/A	N/A	1.000	1.000	1.000
	**Dorsal side**	HD1	**<0.001**	0.255	0.075	N/A	N/A	N/A	0.405	1.000	1.000
**Sweat accumulation**		N/A	**<0.001**	0.607	**<0.001**	N/A	N/A	N/A	**<0.001**	**<0.001**	1.000

A: barefoot, B: leather sports shoes, C: mesh sports shoes, and D: closed-toe slippers. Note: Variables with *p*-value in bold are significant (*p* < 0.05); N/A: not applicable.

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
