# Peer review of "Influence of Upper Footwear Material Properties on Foot Skin Temperature, Humidity and Perceived Comfort of Older Individuals"

_ijerph, 2022, doi:10.3390/ijerph191710861_

Round 1

Reviewer 1 Report

In this manuscript, the authors examine the effects of upper footwear materials on changes and distributions in the foot skin temperature and relative humidity for older individuals. Here are the comments and suggestions:

1. The author examined the effects of upper footwear materials on changes and distributions in the foot skin temperature and relative humidity for older individuals. But there are lots of similar studies before. In addition, it is common sense that fabric materials with large air holes showed greater gas permeability than other materials such as synthetic leather and composite layers. So what is the novelty of this study?

2. The shape of the upper and the materials of the soles of the three shoes are different. Why did the author design experiment in this way? Did the author consider the confounding influence of the above factors on the wearing comfort results?

3. The schematic experimental procedures (Figure 2) need to be redrawn more clearly and simply.

4. In the 1.1 statistical analysis part, the versions of MATLAB and SPSS software need to be updated. The prerequisite for using Spearman's correlation coefficient is to test the normality of the data. Has the author tested the normality of the data? Please state in the text.

5. All the numbers before the heading are wrong. Please check very carefully.

6. The templates in Figure 9 were put in the wrong place. Please check all the figures and tables and make sure no mistakes happen again.

7. The English of the overall manuscript should be improved.

Reviewer 2 Report

At first glance, this is a text that already impresses for its quality and scientific rigor and that should be considered for publication in the prestigious IJERPH following the guidelines below.

The fellow researchers from Hong Kong brought an innovative study that very properly shows their results through thermographic evaluation, which demonstrates a great deal of mastery of the analysis technique.

It is an essential and unprecedented study on thermal comfort of the feet and that is concerned with the health of the elderly population, but this health aspect should be emphasized more in the text. There do not seem to be many similar studies yet in the literature in this regard. Therefore, this work can be cited by many other studies that will follow.

Abstract

1. Add numerical values ​​from the results to the summary, this is be further explained below.

Keywords

2. Add thermography

Introduction.

It's well-founded. It shows knowledge of the subject and concern with thermoregulation in the elderly.

3. The importance of this study for human health remains to be better explained.

4. Need to make it clearer which age you are considering as elderly in the text. Over 60 years?

Objective

5. Inform which materials you are comparing, i.e., specify better what makes the shoes different from each other. Are you comparing sneakers with higher and lower thermal insulation index?

6. I think it's better to remove the phrase that comments “various activities.” The study is very specific for treadmill walking activity as detailed in the method, so it is not evaluating various recreational or sports activities. Specify better what you are considering as “various activities” in the subjective sensation or withdraw from the objective. The objective must be well aligned with the conclusion, and with a clear question and answer.

Participants

7. Inform if any participant had skin disease, pre-diabetes, wounds, varicose veins, ankle edema, or foot hyperhidrosis.

8. Inform if the level of cognition and verbal comprehension was within the normal range, since this is a study with a subjective questionnaire in the elderly.

9. Did the volunteers have any regular physical conditioning or were they sedentary?

10. Was the clinical evaluation of the volunteers carried out by a physician or simply a questionnaire?

Experimental conditions.

11. Figure 1 is quite self-explanatory and the text makes it clear what motivated the authors to define the groups and choose these materials. However, making this choice clearer in the purpose of the study.

12. Did the authors already have experience and publication with any type of these footwear groups B, C and D? Cite in the reference if applicable.

Experimental protocol

In order for the study to be reproduced by other researchers, it would be important to clarify the following points that were left open:

13. It would be interesting to add which guidelines were used for thermographic assessment. Looks like American Academy of Thermology guidelines to me (see reference SCHWARTZ,2021)

14. How was air velocity measured to confirm passive convection loss <0.2 m/s?

15. Were all subjects together in the same room or was it an individual study? Room temperature may vary with a larger number of people inside.

16. Confirm that the volunteers were still when SITTING or engaged in any physical activity at this time. If yes, in what quantity, frequency and intensity?

17. Please specify which black anti-reflective fabric was used. Anti-glare to visible or infrared radiation?

18. Correct. “Acclimatization,” despite being a very common term in publications, is more correctly used for more prolonged situations adaptive to the climate (mountains, for example). I suggest changing to the word “thermalization,” which is the most appropriate term for fast thermal conditioning situations, less than 1 hour.

19. Did barefoot volunteers step on a thermally insulated floor? What was the insulating material of the floor?

20. I was very concerned about varying the time from 20 to 40 min after exercise and using cold water and alcohol to cool my feet. Were the feet completely immersed or sprayed with coolant only on the back or soles? It seems to me a bias that should be analyzed more carefully in future studies, as differences of 20 min and artificial cooling with alcohol can induce very different thermoregulatory responses between individuals and even a reactive hyperemia response. How long on average did recovery take to the early stage? Inform in the text.

Statistical analysis

21. Did everyone have the right foot as dominant?

22. The temperature measurement on the graph corresponded to a single point on the image (temperature line pixel) rather than an average temperature of a ROI (eg range). The choice of a line with a single temperature point should be considered a small bias in the study since it is limited to a very small area and not to a region of the foot. Note that the result of this study was obtained through the analysis of a thermal profile (thermal line and not ROI). The presence of superficial veins on the dorsum of the foot is common, so the comparison of a point that falls on a venous region will have a higher temperature than that of a region without a superficial vein (this could explain a higher standard deviation in certain regions).

23. How was it ensured that the choice of the location of the point (thermal line) of temperature measurement was the same in all images since the feet vary anatomically? Was it done manually?

24. How many temperature points were evaluated in total? Explain better how the graph of figure 5 was constructed for all feet.

25. Figure 4. Explain in the legend that the first figure with a line marking was used to measure the temperature in the percentage in thermography.

Results

Changes and distribution of skin temperature

26. Dorsal D1 and dorsal D1 appeared to be different points. If yes, then rename different so as not confuse the reader.

Figure 5.

27. Inform in the caption what A, B, C, and D are. The caption must be explanatory, regardless of reading the entire article.

28. Add the color scale next to the figures, which can be individually if the minimum and maximum temperatures are different, or a single one representing all the images. Ideally, all images should be in the same temperature range (thermal window). The reader must have an idea of ​​the temperature that each color represents and the comparison has to be with images under the same thermal window.

29. Does the graphic really represent all the volunteers, or is it an illustration of four volunteers one from each group?

Figure 6.

30. Inform in the caption what A, B, C, and D are. The caption must be explanatory, regardless of reading the entire article.

31. I was curious that the study also presents a Figure 6.1 showing the thermal differential as in Figure 6, but comparing it with condition A (natural barefoot). That is, how much are conditions B, C, and D different from condition A (natural barefoot). This would provide a new perspective to the reader of the result and a comparison of what the thermal comfort of the sneakers is in relation to a natural bare foot. Does this make sense to the authors?

Table 2.

32. Inform in the legend the anatomical location of points D and P. Avoid using the same point name for different locations, eg dorsal D1.

Figure 9.

33. Inform in the caption what A, B, C, and D are. The graphic formatting could be revised to make it visually better.

34. Has BMI, age, sex, humidity, sweating, or foot size been related to higher or lower temperature of the feet? Inform this better in the text or table.

Discussion.

35. Start with the most impactful sentence, highlighting the main finding and not stating the purpose of the study again. That is, take off the first sentence and start with the second.

36. Quantitatively inform how much was the temperature increase with the use of shoes in relation to barefoot. Inform if there is anything in the literature that found similar or different results from the findings.

37. 4º C maximum change was quoted, add standard deviation.

38. Inform the values ​​found by Shimazaki and Murata (2015) for the reader to have a quantitative idea of ​​the result. And what is the methodological difference between this study and that of Shimazaki and Murata (2015)?

39. Did the volunteers' weight influence the thermal results? Explain better the comparison with references 9 and 34 cited.

40. To better explain the comparative result between the dorsum and soles of the feet in the different groups. Explore and add a dorsum/plantar index may be interesting (including in a table or graph). This can be an interesting thermal comfort index.

41. Comment on whether the fact of tying the shoelaces with greater pressure in shoes B and C can influence the result, for example, altering the thermal conduction and venous flow of the dorsum of the foot.

42. Explain how wearing socks would change the results of this study? Would the same thermal proportions be maintained between the groups?

43. There seems to be a limitation in this study, which is the lack of previous baropodometry examination to know if the volunteers really had a normal gait.

44. Why did the Footwear B condition show a slightly higher temperature distribution from the instep to the dorsal and plantar heels? It is described in the results part, but there is nothing in the discussion on it.

45. As it is a Public Health journal, informs what risks B and D shoes can bring to the health of the elderly. Whether this could cause any harm to diabetic feet.

Conclusion

46. ​​Make it clear that condition C shoes are the most recommended for seniors. Do the same in the summary, highlighting the importance of the C shoe. The conclusion is still not good, it was necessary to add the most important quantitative data found by the authors to clarify the magnitude of the findings. Complement the adjectives “very,” “bigger”, “higher”, “easily” with quantitative data, but emphasizing condition C, which is related to better foot health. This will also facilitate the understanding and motivation of readers and other researchers so that they can replicate and improve the results obtained from this study.

Once again, I emphasize my admiration for the scientific rigor of the researchers and the care taken to arrive at these results that deserve to be published in this journal, following the numerous but small revisions to be corrected given above.

References:

SCHWARTZ, Robert G. et al. The American Academy of Thermology Guidelines for Breast Thermology 2021. Pan American Journal of Medical Thermology, [S.l.], v. 8, p. 003, dez. 2021. ISSN 2358-4696.

Balbinot LF, et al. Repeatability of infrared plantar thermography in diabetes patients: a pilot study. J Diabetes Sci Technol. 2013 Sep 1;7(5):1130-7.
